# Transcriptome Analysis Reveals the Response Mechanism of *Digitaria sanguinalis*, *Arabidopsis thaliana* and *Poa annua* under 4,8-Dihydroxy-1-tetralone Treatment

**DOI:** 10.3390/plants12142728

**Published:** 2023-07-22

**Authors:** Qiumin Sun, Tao Wang, Jiu Huang, Xinyi Gu, Yanling Dong, Ying Yang, Xiaowen Da, Xiaorong Mo, Xiaoting Xie, Hangjin Jiang, Daoliang Yan, Bingsong Zheng, Yi He

**Affiliations:** 1State Key Laboratory of Subtropical Silviculture, Zhejiang A&F University, Hangzhou 311300, China; sunqiumin99@163.com (Q.S.); wangtao042500@163.com (T.W.); guxinyi200715@163.com (X.G.); 17867963651@163.com (Y.D.); sky_yangying@163.com (Y.Y.); xiexiaoting@stu.zafu.edu.cn (X.X.); liangsie@zafu.edu.cn (D.Y.); bszheng@zafu.edu.cn (B.Z.); 2Zhejiang Provincial Key Laboratory of Forest Aromatic Plants-based Healthcare Functions, Zhejiang A&F University, Hangzhou 311300, China; 3School of Environment science and Spatial Informatics, China University of Mining and Technology, Xuzhou 221116, China; jhuang@cumt.edu.cn; 4State Key Laboratory of Plant Physiology and Biochemistry, College of Life Science, Zhejiang University, Hangzhou 310058, China; dxwpp@zju.edu.cn (X.D.); xiaorong@zju.edu.cn (X.M.); 5Center for Data Science, Zhejiang University, Hangzhou 310058, China; jianghj@zju.edu.cn

**Keywords:** 4,8-DHT, photosynthesis, antioxidant enzymes, herbicides

## Abstract

4,8-dihydroxy-l-tetralone (4,8-DHT) is an allelochemical isolated from the outer bark of *Carya cathayensis* that acts as a plant growth inhibitor. In order to explore the mechanism of 4,8-DHT inhibiting weed activity, we treated three species of *Digitaria sanguinalis*, *Arabidopsis thaliana,* and *Poa annua* with different concentrations of 4,8-DHT and performed phenotype observation and transcriptome sequencing. The results showed that with an increase in 4,8-DHT concentration, the degree of plant damage gradually deepened. Under the same concentration of 4,8-DHT, the damage degree of leaves and roots of *Digitaria sanguinalis* was the greatest, followed by *Arabidopsis thaliana*, while *Poa annua* had the least damage, and the leaves turned slightly yellow. Transcriptome data showed that 24536, 9913, and 1662 differentially expressed genes (DEGs) were identified in *Digitaria sanguinalis*, *Arabidopsis thaliana,* and *Poa annua*, respectively. These DEGs were significantly enriched in photosynthesis, carbon fixation, glutathione metabolism, phenylpropanoid biosynthesis, and oxidative phosphorylation pathways. In addition, DEGs were also enriched in plant hormone signal transduction and the MAPK signal pathway in *Arabidopsis thaliana*. Further analysis showed that after 4,8-DHT treatment, the transcript levels of photosynthesis PSI- and PSII-related genes, LHCA/B-related genes, Rubisco, and PEPC were significantly decreased in *Digitaria sanguinalis* and *Arabidopsis thaliana*. At the same time, the transcription levels of genes related to glutathione metabolism and the phenylpropanoid biosynthesis pathway in *Digitaria sanguinalis* were also significantly decreased. However, the expression of these genes was upregulated in *Arabidopsis thaliana* and *Poa annua*. These indicated that 4,8-DHT affected the growth of the three plants through different physiological pathways, and then played a role in inhibiting plant growth. Simultaneously, the extent to which plants were affected depended on the tested plants and the content of 4,8-DHT. The identification of weed genes that respond to 4,8-DHT has helped us to further understand the inhibition of plant growth by allelochemicals and has provided a scientific basis for the development of allelochemicals as herbicides.

## 1. Introduction

Herbicides are the most common ways to control weeds, of which chemical herbicides are the main ones used today, but their long-term use leads to weed resistance, which weakens the inhibitory effect of herbicides on weeds and can also cause a number of environmental problems [1,2,3]. It is believed that biological or natural product herbicides with novel mechanisms of action will give life to the development of herbicides. Plants secrete various metabolites such as water-soluble organic acids, quinones, phenolic compounds, cinnamic acids, terpenoids, and steroids, which inhibit weed seed germination and growth without damaging the crop and can be rapidly degraded in the background; these metabolites are known as allelochemicals [4]. For example, the flavone linarin, a compound proposed from *Zanthoxylum affine*, was able to inhibit the germination and residual growth (root and stem elongation) of *Lactuca sativa* (lettuce) and *Lolium perenne* (perennial ryegrass) [5]. Sarmentine is a substance isolated from long pepper (*Piper longum* L.) fruits that inhibits the activity of redroot pigweed, barnyardgrass, bindweed (*Convonvulus* sp.), and hairy crabgrass and has been applied as an herbicide [6]. Thus, plant-derived allelochemicals have high research value and broad application prospects.

When plants are stressed, the various organs and metabolic processes of the plant are disrupted, such as oxidative phosphorylation in mitochondria [7], photosynthesis in chloroplasts [8], endoplasmic reticulum stress [9], and the formation of various secondary metabolites [10]. Simultaneously, in response to ROS production due to oxidative stress and to regulate cellular redox homeostasis [11,12], plants utilize a multifaceted and robust antioxidant defense system in which nonenzymatic and enzymatic components play a role in sensing and eliminating excess ROS [13]. Furthermore, phytohormones and MAPK-mediated signaling pathways also play a key role in the adaptive growth of plants as a result of environmental stimuli [14,15]. For instance, abscisic acid (ABA) is a signaling molecule and key regulator of plant stress responses [16,17].

4,8-DHT is an allelochemical extracted from the outer bark of *Carya cathayensis* and has been shown to have an inhibitory effect on plant growth [18,19]. The molecular structures are shown in Figure 1A. Accumulating evidence suggests that 4,8-DHT inhibits the germination of lettuce (*Latuca sativa* L.) and cucumber (*Cucumis sativus* L.), and reduces the germ length, radicle length, and fresh weight of onion (*Allium cepa* L.) seedlings and lettuce [18]. Another study argues the intensity of phytotoxicity depends on the tested plants and the amount of 4,8-DHT [20]. It was discovered that 4,8-DHT and its five derivatives all exhibit significant herbicidal activity towards the seed germination and seedling growth of six representative weeds [21]. At the same time, this report pointed out that high concentrations of 4,8-DHT inhibit weed growth, whereas low concentrations promote weed growth. These show that the phytotoxicity of 4,8-DHT has dosage, action target (plant type), and content (seed germination or seedling growth) selectivity.

It has been reported that *Digitaria sanguinalis* and *Poa annua* are annual grasses with a widespread distribution and are considered among the most challenging weeds to control globally. These weeds have developed multiple resistances due to the long-term use of herbicides. *Poa annua* is ranked as the most resistant weed by number of loci, having developed resistance to 12 different herbicide loci worldwide [22]. For example, *Poa annua* has shown resistance to herbicides targeting ALS and photosystem II [23]. *Digitaria sanguinalis* has evolved four modes of action for herbicide resistance, including acetolactate synthase (ALS), acetyl coenzyme A carboxylase (ACCase), 5-enolpyruvylshikimate 3-phosphate synthase (EPSPS) and photosystem II [24]. *Arabidopsis thaliana* as a model plant has also produced resistance loci, such as ALS, which have been used several times to study herbicide trials [25]. Therefore, finding new herbicides that can effectively target these weeds is of utmost importance.

Although 4,8-DHT has been shown to inhibit the activity of weeds by physiological indicators [20,21], the intrinsic mechanism of its inhibitory effect on weeds is still unclear. In this study, the transcriptome sequencing of *Digitaria sanguinalis, Poa annua,* and a model plant, *Arabidopsis thaliana,* was performed to identify some key regulatory pathways and gene families induced after the application of 5 mM 4,8-DHT at 0 h, 2 h, and 2 d, respectively. This study not only reveals the inhibitory effect of 4,8-DHT on weeds, but also the different levels of response of different species to 4,8-DHT, and it also provides new insights into the mechanisms of plant growth inhibition by allelochemicals.

## 2. Results

### 2.1. The Effect of 4,8-DHT on the Growth of Three Plant Species

To accurately evaluate the effects of different concentrations of 4,8-DHT on plants, three species were tested. We observed the morphological appearance and measured the fresh weight of *Digitaria sanguinalis*, *Arabidopsis thaliana,* and *Poa annua* treated with different concentrations of 4,8-DHT (Figure 1B and Appendix A). The results reveal that the damage degree of plants progressively worsened with the increase in the 4,8-DHT concentration. Specifically, the leaves of the plants treated with 5 mM 4,8-DHT exhibited signs of wilting and curling, whereas some plants did not survive under the treatment of 15 mM 4,8-DHT. Whereas the plant fresh weight was significantly lower under the 5 mM and 15 mM 4,8-DHT treatments compared to CK. In addition, the morphological response to the same concentration of 4,8-DHT treatment varied among *Digitaria sanguinalis*, *Arabidopsis thaliana,* and *Poa annua*. Among these, *Digitaria sanguinalis* sustained the greatest damage, with leaves shrinking and aging, and roots darkening progressively. *Arabidopsis thaliana* was the second most damaged, with leaf shrinkage, while *Poa annua* showed the least damage, showing a slight yellowing of the leaves (Figure 1B and Appendix A). Thus, in agreement with previous studies [20], the extent to which plants were affected depended on the tested plants and the amount of 4,8-DHT.

### 2.2. Analysis of Differentially Expressed Genes after the 4,8-DHT Treatment

To characterize transcriptional variations occurring in the three species after 4,8-DHT treatment, we sequenced the transcriptomes of seedlings treated with 5 mM 4,8-DHT. Based on transcriptome data, a differential gene expression analysis was performed. Differentially expressed genes (DEGs) were screened using a *q* value < 0.05 and |log_2_(Fold change)| ≥ 1 as criteria. The results show that there were significant variations in the DEGs of *Digitaria sanguinalis*, *Arabidopsis thaliana,* and *Poa annua* under the 4,8-DHT treatments. Compared with *Arabidopsis thaliana* and *Poa annua*, the number of DEGs of *Digitaria sanguinalis* was the highest 0 h (CK), 2 h, and 2 d after the treatment of 4,8-DHT, which were 7217, 4199, and 6486, respectively (Figure 2A and Appendix A). Especially after 2 h of treatment, 6528 genes in *Digitaria sanguinalis* were downregulated, indicating that *Digitaria sanguinalis* had a more severe response 2 h after the treatment with 4,8-DHT and was more susceptible to it than the other two species. It is worth noting that the number of *Poa annua* DEGs was relatively low, reaching 352, 609, and 701 at 0 h, 2 h, and 2 d, respectively, and upregulated DEGs were continuously more numerous than downregulated DEGs, demonstrating that, consistent with the phenotypic evidence, *Poa annua* may be less susceptible to response than the other two species. A comprehensive analysis based on the Venn diagram showed that only 1 gene was consistently upregulated and 1 gene was consistently downregulated in *Poa annua*, 52 genes were consistently upregulated and 10 genes were consistently downregulated in *Digitaria sanguinalis*, and 27 genes were consistently upregulated and 17 genes were consistently downregulated in *Arabidopsis thaliana* at 0 h, 2 h, and 2 d (Figure 2B). Thus, these genes may be key genes in the response to 4,8-DHT treatment.

### 2.3. GO Classification and KEGG Analysis of Differentially Expressed Genes

To reveal the response mechanism of *Digitaria sanguinalis, Arabidopsis thaliana, and Poa annua* after the treatment of 4,8-DHT, KEGG pathway and GO enrichment analyses were performed to categorize the DEGs. The analysis results revealed that DEGs in *Digitaria sanguinalis, Arabidopsis thaliana, and Poa annua* were classified into different GO terms, including three categories: “biological process”, “cellular component”, and “molecular function”. In *Digitaria sanguinalis*, the main terms in the biological process category were “regulation of transcription, DNA-templated”, “oxidation-reduction process”, and “transcription, DNA-templated”. The main enriched GO terms for the cellular components category were “nucleus”, “cytoplasm”, and “chloroplast”. The main terms for the molecular function category were “protein binding”, “ATP binding”, and “DNA binding” (Appendix A). The main terms for the biological process category and the molecular function category in *Arabidopsis thaliana* were the same as in *Digitaria sanguinalis*; the main enriched GO terms for the cellular component category were “nucleus”, “integral component of membrane”, and “chloroplast” (Appendix A). In *Poa annua*, the main terms in the biological processes category were “protein phosphorylation”, “lipid transport”, and “response to oxidative stress”. The main enriched GO terms for the cellular component category were “plasma membrane”, “integral component of membrane”, and “extracellular region”. The main terms for the molecular function category were “protein binding”, “ATP binding”, and “kinase activity” (Appendix A). In summary, the enrichment processes during the 4,8-DHT treatment did not differ significantly between the three species, with all undergoing redox reactions and cellular activity in response to the stress.

Then, we mapped DEGs to the reference canonical pathways in the KEGG database. The results clearly stated that in the 2 h vs. CK and 2 d vs. CK groups of *Digitaria sanguinalis* and *Arabidopsis thaliana*, DEGs were mainly enriched in “photosynthesis”, “photosynthesis-antenna proteins”, “carbon fixation in photosynthetic organisms”, “glutathione metabolism”, and “phenylpropanoid biosynthesis”, which suggested that they may have partly the same response pathway to the treatment of 4,8-DHT (Figure 3A,B). The DEGs in the 2 d vs. CK group of *Digitaria sanguinalis* were also enriched in two significant pathways: “protein processing in endoplasmic reticulum” and “oxidative phosphorylation” (Figure 3A). Moreover, the DEGs were enriched in “Plant hormone signal transduction” and “MAPK signaling pathway-plant” in *Arabidopsis thaliana* (Figure 3B). In addition, the DEGs in the 2 h vs. CK and 2 d vs. CK groups of *Poa annua* were mainly enriched in “ribosome”, “glutathione metabolism”, “phenylpropanoid biosynthesis”, and “oxidative phosphorylation” (Figure 3C). Taken together, these results suggest that the treatment of 4,8-DHT influences energy metabolism such as photosynthesis and oxidative phosphorylation, signal transduction such as MAPK and hormones, the synthesis of secondary metabolites such as phenylpropanoids, and antioxidants such as glutathione in response to stress, and that the pathways significantly enriched by 4,8-DHT differ between species. The analysis of these KEGG pathways will help to understand how 4,8-DHT inhibits plant growth.

### 2.4. The Identification of Key Pathways’ Response to the 4,8-DHT Treatment

To further understand the response of critical pathway-related genes to 4,8-DHT treatment, the expression profiles of these genes in the three species were studied using hierarchical clustering to find gene clusters with similar expression patterns. Each pathway displayed a distinct temporal expression pattern, with colored modules representing the most significant of these DEGs, and subsequent DEGs analyses focused on these colored significant modules. Glutathione metabolism [26] and phenylpropanoid biosynthesis [27] are important pathways for controlling ROS in abiotic stresses. Therefore, we analyzed significant DEGs in the glutathione metabolism and phenylpropanoid biosynthesis pathways in three species (Figure 4). In *Arabidopsis thaliana*, the glutathione metabolism and phenylpropanoid biosynthesis pathways had the largest number of DEGs, and the expression of most of the genes was upregulated after 2 h of treatment. At the same time, the number of DEGs in the glutathione metabolism pathway in *Poa annua* was small, but all of them were significantly upregulated at the later stage of treatment, showing that *Poa annua* may be less stressed and produce less antioxidant enzymes. However, the opposite trend occurred in *Digitaria sanguinalis*. The expression of most glutathione metabolism and phenylpropanoid-biosynthesis-related genes was significantly downregulated at 2 h, while a small number of genes were upregulated in glutathione metabolism. This result indicated that the 4,8-DHT treatment inflicted significant damage on the plants, hindered the production of antioxidant enzymes, and boosted the expression of genes associated with ROS formation, which could lead to an excess of ROS in plants and oxidative damage to plants.

Moreover, the STEM red module plots show that the three photosynthesis-related pathway-related genes of *Digitaria sanguinalis* were continuously downregulated with the extension of 4,8-DHT treatment time (Figure 5). In *Arabidopsis thaliana*, the expression levels of genes related to photosynthesis pathways decreased after 2 h of treatment of 4,8-DHT. However, there was an increase in the expression of these genes at 2 d compared to the 2 h of 4,8-DHT treatment (Figure 5). It is indicated that *Digitaria sanguinalis* was continuously downregulated by 4,8-DHT, whereas *Arabidopsis thaliana* slowly underwent repair at 2 d. The presence of antioxidant enzymes is conceivable. Surprisingly, the DEGs in *Poa annua* were not enriched in photosynthesis-related pathways, and 4,8-DHT did not appear to alter its photosynthetic system. However, 4,8-DHT treatment had an effect on the oxidative phosphorylation pathway of *Poa annua*, with the expression of associated genes increasing significantly at 0 h of treatment and then decreasing (Appendix A). *Poa annua* can react to oxidative damage caused by 4,8-DHT, primarily through the oxidative phosphorylation process. Simultaneously, genes associated with the oxidative phosphorylation pathway were also affected in *Digitaria sanguinalis*, but most of them were significantly increased in the subsequent phase of treatment (Appendix A). Furthermore, *Arabidopsis thaliana* has a substantial number of DEGs in MAPK and hormone response pathways (Appendix A). The DEGs of the plant hormone signal transduction and MAPK signaling pathways were substantially upregulated following the treatment of 4,8-DHT at 2 h and downregulated following the treatment of 4,8-DHT at 2 d, as shown in the STEM significant module plot. These findings demonstrate that various species respond to 4,8-DHT treatment via different physiological mechanisms in response to stress, increasing resistance.

### 2.5. The Identification of Key 4,8-DHT-Responsive Genes

To understand which key genes were involved in the response to 4,8-DHT, a correlation network analysis of DEGs from the STEM significant module was performed in the three species. Of these nine KEGG pathways analyzed, 64, 34, and 12 high-link genes were selected as key genes in *Digitaria sanguinalis*, *Arabidopsis thaliana,* and *Poa annua*, respectively (Appendix A). Glutathione is an essential metabolite in plant life and has a major role in controlling reactive ROS [26]. In our study, the glutathione-related genes *GSR*, *GPX6*, and *GSTU6* were upregulated in *Arabidopsis thaliana* and *Poa annua*, respectively, after 2 h of treatment with 4,8-DHT, and may play a crucial role in antioxidation (Figure 6B,C and Figure 7). However, the GST (*GSTT1* and *GSTU6*), *GPX*, and NAD(P)H-quinone oxidoreductase subunit L (*NDHL*) genes were downregulated in *Digitaria sanguinalis*, which could be related to 4,8-DHT causing severe damage and limiting the synthesis of antioxidant enzymes (Figure 6A and Figure 7A). POD is an antioxidant enzyme involved in the production of phenylpropane [28]. Increased POD activity eliminates excess ROS and free radicals from plants, minimizing the damage caused by stressed plant tissues. *Digitaria sanguinalis*, *Arabidopsis thaliana,* and *Poa annua* all had only one key gene, *PODA2*, *POD*, and *POD43*, respectively (Figure 7). This suggests that antioxidant key genes in *Arabidopsis thaliana* and *Poa annua* were induced to be expressed by 4,8-DHT to scavenge ROS. Some allelochemicals are well recognized photosynthesis inhibitors, owing to their destructive action against chlorophylls and other photosynthetic pigments or the disruption of electron transport chains [29]. In our study, a total of 15 key photosynthesis-related genes were identified in *Digitaria sanguinalis* with a downregulated trend for 2 h of treatment, including PSI (*PsaD*, *PsaG*, and *PsaH*), PSII (*Psb28* and *PsbY*), and ATP synthase (*ATPF1G* and *ATPG*) in the photosynthetic pathway; the photosynthesis antenna proteins of LHC (*LHCA1*, *LHCA4*, *LHCB4* and *LHCB7*); and carbon fixation in photosynthetic organisms by *Rubisco*, glyceraldehyde-3-phosphate dehydrogenase A (*GAPA*), and phosphoenolpyruvate carboxylase (*PEPC*) (Figure 7A, Figure 8A and Appendix A). Simultaneously, in *Arabidopsis thaliana*, a total of 23 photosynthesis-related key genes were downregulated after 2 h of treatment, including PSI (*PsaD*), PSII (*PsbQ*, *PPL*, *Psb27*, and *Psb28*), ATPase (*ATPC1* and *ATPF0B*), and ferredoxin-NADP-oxidoreductase (*FNR*), LHC for photosynthesis antenna proteins (*LHCB3*, *LHCB4.2*, and *LHCB6*), and carbon fixation in photosynthetic organisms by *Rubisco* (Figure 7,Figure 8B and Appendix A). It is possible that these genes severely impede biomass accumulation and the growth and development of the recipient plant. In addition, ABA-related genes, such as abscisic -acid-responsive element-binding factor 3 (*ABF3*), protein phosphatase 2C (*PP2C*), and hypersensitive to ABA1 (*HAB1*), and salicylic acid (SA) receptors (*NPR3*) were identified in *Arabidopsis thaliana,* and these genes were significantly upregulated after 2 h of treatment with 4,8-DHT (Figure 7B and Appendix A). There were also key genes identified in the process of oxidative phosphorylation, such as cytochrome c oxidase subunit (*COX*), ATP synthase subunit, NADH-ubiquinone oxidoreductase subunit (*NDUF*), etc. (Figure 7 and Appendix A). These genes had the opposite trends in *Digitaria sanguinalis* and *Poa annua*, with upregulation after 2 d of treatment in *Digitaria sanguinalis* and downregulation after 2 h of treatment in *Poa annua*. It is suggested that the key genes that respond to 4,8-DHT differ among species and that trends vary. These essential genes collectively contribute significantly to the response to 4,8-DHT in all three species and will be further analyzed.

## 3. Discussion

### 3.1. The Downregulation of Photosynthesis-Related Genes in Response to Treatment with 4,8-DHT

Photosynthesis is essentially one of the key plant processes that directly determines crop productivity. The reduced productivity of many plants in harsh environmental conditions is often associated with reduced photosynthetic capacity [8]. Some chemosensory chemicals, such as juglone, sorgoleone, thymol, and chalcone, are photosynthesis inhibitors, mainly due to their destructive activity towards chlorophyll and other photosynthetic pigments, disruption of the electron transport chain, and also disruption of the carbon fixation pathway [26]. In *Digitaria sanguinalis* and *Arabidopsis thaliana*, genes encoding proteins in PSI (*PsaD*, *PsaG*, and *PsaH*), PSII (*Psb28*, *PsbY*, *PsbQ*, and *PPL1*), ATP synthase (*ATPF1G*, *ATPG*, *ATPC1*, and *ATPF0B*), and several components of the photosystem-associated light-harvesting complex (*LHCA* and *LHCB*) were downregulated after Among them, PsbQ and PsbP (a homolog of PPL1) are extrinsic proteins of photosystem II and component subunits of the oxygen-evolving complex (OEC), which is involved in oxygen release [30]. The downregulation of *PsbQ* and *PsbP* may result in defective OEC function and a lack of electron supply, which may disrupt the PSII reaction center. An unstable OEC also causes the formation of ROS [31]. Simultaneously, *PPL1* (a homolog of *PsbP*) also plays an important role in the repair of PSII damage [32,33]. In *Arabidopsis thaliana,* PPL1 deletion mutants were discovered, as well as a decrease in PSII core protein D1 accumulation and a delay in PSII activity recovery after photoinhibition [34]. The PSII subunits Psb28 and PsbY also protect PSII from light-induced damage [35]. Therefore, when these PSII components are damaged, ROS production is accelerated and PSII repair is delayed. In addition, the photosystem-associated light-harvesting complex mainly harvests light energy and transmits it to photosynthetic reaction centers; the chlorophyll a/b binding proteins (*LHCA* and *LHCB*) are also downregulated when plants are subjected to abiotic stresses, which is consistent with our findings [36]. This shows that 4,8-DHT treatment may inhibit light energy absorption, transmission, and dispersion. PSI will be far more stable and less likely to be destroyed than PSII, which is typically produced by damage to the structure of the PSII-LHCII complex. Under stress, the phosphorylation of PSII proteins is much faster, and the intact structure of the PSII-LHCII supercomplex is disrupted, generating an infinite transfer of excess excitation energy from PSII-LHCII to PSI. When the electron supply of PSII exceeds the electron-receiving capacity of PSI, this causes damage to PSI [37,38]. The downregulation of *PsaD*, *PsaG*, and *PsaH* genes may lead to a decrease in PSI activity, which in turn hinders electron transfer from PSII to PSI, increases excess excitation energy, leads to an increase in ROS, and intensifies the damage to PSII.

Moreover, carbon fixation is a crucial part of photosynthesis, and it has been shown in various stressors that stress limits photosynthesis by reducing Rubisco content and activity [39,40]. *Arabidopsis thaliana* and *Poa annua* are C3 plants that fix carbon via Rubisco [41]. *Digitaria sanguinalis* is a C4 plant that usually fixes carbon via PEPC and Rubisco. Under drought conditions, PEPC has a higher capacity to fix CO_2_ than Rubisco [42,43]. In our study, carbon-fixation-related genes (*Rubisco*, *ALT2*, and *PEPC*) were also affected, with carbon-fixation-related genes being downregulated after 2 h and 2 d of treatment with 4,8-DHT. Taken together, the downregulation of *Rubisco* and *PEPC* in the carbon fixation pathway may be one of the reasons why *Digitaria sanguinalis* is more severely stressed than that in *Arabidopsis thaliana*. Our findings also demonstrated that 4,8-DHT treatment reduced the expression of all essential genes involved in the photosynthetic process. Thus, the impact on photosynthesis in plants and the decrease in plant yield may be the primary mechanism behind the herbicidal effect of 4,8-DHT. It can also be speculated that more photosynthesis-related genes were downregulated in *Digitaria sanguinalis* and *Arabidopsis thaliana* compared with *Poa annua*, which may be the main reason for their more severe damage. In this study, it can only be shown that important components of photosynthesis are downregulated by 4,8-DHT, but exactly how 4,8-DHT disrupts photosynthesis needs to be further explored and verified at a later stage in order to identify the target site of action of 4,8-DHT and improve weed control efficiency.

### 3.2. ROS Scavenging System May Be the Key to the Response to 4,8-DHT

Numerous studies have shown that plants dealing with abiotic stresses activate both enzymatic and nonenzymatic antioxidant mechanisms to eliminate reactive oxygen species (ROS), mitigate ROS-induced harm to plant cells, and enhance plant antioxidant ability [44,45]. POD, GST, GPX, and other enzymatic components are induced in response to abiotic stresses. For example, the overexpression of *OsGSTU6* in rice reduced Cd accumulation in leaves and enhanced plant tolerance to Cd stress, while knockout rice mutants of *OsGSTU6* increased Cd accumulation in rice leaves and reduced stress tolerance [46]. Cucumber’s GPX4 gene is upregulated in response to cold stress, producing resistance [47]. Notably, both the glutathione and phenylpropane pathways showed sensitivity to 4,8-DHT in *Digitaria sanguinalis*. However, in *Arabidopsis thaliana* and *Poa annua*, the expression patterns were distinct. The key genes *GSR*, *GPX6*, and *POD* in *Arabidopsis thaliana* and *GSTU6* and *POD43* in *Poa annua* were upregulated after 2 h of treatment with 4,8-DHT, whereas the key genes GST (*GSTT1* and *GSTU6*), *GPX, NDHL,* and *PODA2* in *Digitaria sanguinalis* were downregulated after 2 h of treatment. Other studies on stress have also found that genes responsible for antioxidants show varying patterns of expression in response to abiotic stressors. OsGPXs are increased in response to drought and oxidative stress, whereas they are downregulated in response to salt, heat, and cold [48]. It is speculated that this may be due to the abnormal sensitivity of *Digitaria sanguinalis* to 4,8-DHT, which produces significant quantities of ROS, leading to the inactivation of antioxidant enzymes or its components, exacerbating plant damage. However, no conclusive investigations have been performed to determine the precise mechanism of the downregulation of antioxidant enzymes under stress.

### 3.3. Hormonal Signaling Involved in Resistance Formation after 4,8-DHT Treatment

Photosynthetic-product-related genes in *Arabidopsis thaliana* were considerably downregulated after 2 h of treatment with 4,8-DHT, but the downregulation trend was decreased after the 2 d treatment compared with the 2 h treatment, and some genes began to be upregulated. Hormones may play an important role in explaining this phenomenon. Hormone-related genes in *Arabidopsis thaliana* were upregulated after 2 h of treatment with 4,8-DHT, and recent studies have also demonstrated that plant hormones boost plant stress resistance [49]. In our study, *ABF3*, *PP2C*, and *NPR3* were identified as key genes in the hormone signaling pathway. Under abiotic stress conditions, *ABF3* and *PP2C* are important regulators of ABA signaling that activate antioxidant defense mechanisms in plants [50]. *NPR3*, a receptor for salicylic acid (SA), is also involved in plant resistance to abiotic stress and scavenges ROS [51]. In our study, *ABF3*, *PP2C*, and *NPR3* were upregulated after 2 h of treatment with 4,8-DHT It was reported that the overexpression of *ABF3* in both rice and Arabidopsis has increased plant drought tolerance [50,52]. *PP2Cs* act as coreceptors for ABA signaling, and under dehydration stress, PP2C expression is upregulated to cope with stress and improve plant tolerance [53]. This suggests that *ABF3*, *PP2C*, and *NPR3* play a key role in resistance to 4,8-DHT stress in *Arabidopsis thaliana*.

### 3.4. Oxidative Phosphorylation at Different Expression Levels in Response to 4,8-DHT

The variation in the levels of oxidative phosphorylation between *Digitaria sanguinalis* and *Poa annua* was the focus of our investigation. The PSII repair cycle involves a number of ATP-consuming processes, such as the FtsH degradation of photodamaged D1 proteins and the de novo synthesis of D1 proteins [54]. In the plant body, the two main pathways for ATP production are photosynthesis and respiration. Therefore, a sufficient amount of ATP produced through respiration may be essential for the repair cycle of Photosystem II in situations of impaired photosynthesis [55]. In *Digitaria sanguinalis*, the expression of genes related to ATP synthesis in photosynthesis was downregulated, resulting in an inadequate supply of ATP; thus, the expression of *Digitaria sanguinalis* respiratory-chain-related genes was upregulated to supply ATP to repair damage to PSII, such as *NDUF*, ATP synthesis apparatus subunits, and *COX*. In contrast, the expression of *COX*, *ATP6*, and *ND4* in *Poa annua* was downregulated after 2 h and 2 d of treatment, which may be due to the damage to the respiratory chain. COX catalyzes the transfer of electrons from reduced cytochrome c to the final acceptor of electrons, O_2_, during translocation coupling with H^+^ for ATP production [56]. *COX17* silencing in *Arabidopsis thaliana*, which shows blocked ATP synthesis, also leads to a significant downregulation of other mitochondrial components, including *NDB2* and *ND4* [57]. The plants with reduced *AtCOX17* gene expression exhibit increased levels of ROS following salt treatment, thereby increasing damage to phytoplankton [58]. Therefore, it could also be suggested that different species produce different responses in response to 4,8-DHT treatment.

Here, a fictitious model was created to explain the potential mechanisms of the responses of *Digitaria sanguinalis*, *Arabidopsis thaliana*, and *Poa annua* to the treatment of 4,8-DHT based on the responses of potential candidate genes of these three plants (Figure 9). In this study, the same concentration of 4,8-DHT was used in tests on three different species. Due to the selectivity of the target (plant type) for the phytotoxic activity of 4,8-DHT, the three plants responded to the treatment with 4,8-DHT in various ways. When the plants were stressed, key genes for photosynthesis and carbon fixation were downregulated in response to the 4,8-DHT treatment in *Digitaria sanguinalis* and *Arabidopsis thaliana*, whereas genes in both pathways were not significantly affected in *Poa annua*. Oxidative phosphorylation in *Poa annua* was affected and *COX*, ATP synthase, *ND4* expression was downregulated in response to 4,8-DHT, which produced ROS and affected the growth and development of the plants. In addition, ABA and SA genes *ABF3*, *PP2C*, and *NPR3* were present in *Arabidopsis thaliana* to scavenge ROS and protect the photosynthetic apparatus, thus enhancing plant stress tolerance. Therefore, this suggests that genes related to photosynthesis, antioxidant enzymes, ABA, and SA may play important roles in responding to 4,8-DHT stress and in promoting cell survival.

## 4. Materials and Methods

### 4.1. Plant Material and 4,8-DHT Treatment

The reagent used in this study, 4,8-DHT, was provided by the Ningbo Institute of Technology, Zhejiang University. 4,8-DHT was synthesized according to the synthesis method of Zhang [21]. *Digitaria sanguinalis*, *Arabidopsis thaliana,* and *Poa annua* seeds were purchased from Hangzhou market. In this experiment, three species, *Digitaria sanguinalis* (*Ds*), *Arabidopsis thaliana* (*At*), and *Poa annua* (*Pa*), were chosen as the material for the study. Thirty seeds each of *Digitaria sanguinalis*, *Arabidopsis thaliana,* and *Poa annua* were cultured separately in Petri dishes (9 mm × 15 mm) with 2 layers of already-moistened absorbent paper and filter paper. The seeds were grown under a 12 h light/12 h dark photoperiod (180 μmol m^−2^ s^−1^ light intensity) at 25 °C. Fresh weight and phenotypic observations of 30 randomly selected *Digitaria sanguinalis*, *Arabidopsis thaliana* and *Poa annua* seedlings were measured at 0 h, 2 h and 2 d with 0 mM, 5 mM and 15 mM 4,8-DHT treatments and at least three independent experiments. According to the phenotypic changes and fresh weight measurement, seedlings treated with 5 mM 4,8-DHT were in a state of reduced activity but not death (Figure 1B and Appendix A), so 5 mM 4,8-DHT was selected as the concentration for transcriptome sequencing. Each of the 450 *Digitaria sanguinalis*, *Arabidopsis thaliana,* and *Poa annua* plants were treated with 5 mM 4,8-DHT and water (CK) 7 days after seed germination. A total of 10 seedlings of each species that had reduced activity but were not completely dead were selected for transcriptome analysis after 0 h, 2 h, and 2 d of treatment. For each group, three replicates were established for a total of 27 samples, which were sampled, immediately frozen in liquid nitrogen, and then stored in a −80 °C refrigerator.

### 4.2. RNA Extraction, Library Preparation, and Sequencing

Total RNA was extracted using Trizol reagent (Thermofisher, Waltham, MA, USA, 15596018) from 27 samples (*Ds*: CK1, CK2, CK3, 2 h1, 2 h2, 2 h3, 2 d1, 2 d2, 2 d3; *At*: CK1, CK2, CK3, 2 h1, 2 h2, 2 h3, 2 d1, 2 d2, 2 d3; *Pa*: CK1, CK2, CK3, 2 h1, 2 h2, 2 h3, 2 d1, 2 d2, 2 d3). The Bioanalyzer 2100 and the RNA 6000 Nano LabChip Kit (Agilent, CA, USA, 5067-1511) were used to quantify total RNA amount and purity. The sequencing library was built using high-quality RNA samples with RIN numbers greater than 7.0. The RNA libraries were sequenced by LC Bio Technology CO., Ltd. (Hangzhou, China) on the Illumina NovaseqTM 6000 platform.

### 4.3. Quality Control and Transcriptome Assembly

Using cut-adapt software (https://cutadapt.readthedocs.io/en/stable/, version:cutadapt-1.9, accessed on 3 March 2022), the raw data were stripped of splices; then, the low-quality and duplicate sequences were removed to obtain “Clean” data, which were compared to the genome to obtain the bam file. The low-quality data were then filtered to obtain valid data. The preprocessed valid data were then compared to the reference genome using Hisat2. The mapped reads of each sample were assembled using StringTie (http://ccb.jhu.edu/software/stringtie/, version: stringtie-1.3.4d, accessed on 3 March 2022) with default parameters. Then, all transcriptomes from all samples were merged to reconstruct a comprehensive transcriptome using gffcompare software (http://ccb.jhu.edu/software/stringtie/gffcompare.shtml, version: gffcompare-0.9.8, accessed on 3 March 2022). After the final transcriptome was generated, StringTie and Ballgown (http://www.bioconductor.org/packages/release/bioc/html/ballgown.html, accessed on 3 March 2022) were used to estimate the expression levels of all transcripts and perform expression abundance for mRNAs by calculating the FPKM (fragment per kilobase of transcript per million mapped reads) value.

### 4.4. The Identification and Annotation of Differentially Expressed Genes

Genes differential expression analysis was performed by DESeq2 software (version: DESeq2-3.22.5) between two different groups (and with edge R between two samples). Genes with a false-discovery rate (FDR) of less than 0.05 and an absolute fold change of 2 were considered differentially expressed. Differentially expressed genes were then subjected to an enrichment analysis of GO functions and KEGG pathways. Then, all DEGs were mapped to GO (Gene Ontology) databases (http://www.geneontology.org/, accessed on 3 March 2022) and KEGG (Kyoto Encyclopedia of Genes and Genomes) databases (https://www.genome.jp/kegg/, accessed on 3 March 2022).

### 4.5. Gene Expression Pattern Analysis Using STEM

As there are many genes involved in the key KEGG pathway, to better observe its trend changes and focus on key genes, we used short time-series expression miner (STEM) software (http://www.cs.cmu.edu/~jernst/stem/, version: STEM-1.3.1, accessed on 3 March 2022) to analyze the trends. STEM was used to cluster genes that exhibited a similar temporal expression pattern by using normalized log_2_ ratios of the transcriptomic data, and genes with similar changes at one or more time points were grouped into the same module. This algorithm uses exclusive methods for clustering, comparing, and visualizing data. STEM uses Bonferroni for *p*-value correction, with a change span of 2. The other parameters in the STEM were set to their default values. The STEM was drawn based on R (https://www.r-project.org/, accessed on 3 March 2022) on the OmicStudio platform (https://www.omicstudio.cn/tool, accessed on 3 March 2022).

### 4.6. The Identification of Key Genes via Correlation Network Analysis

In order to identify the key genes of each significant pathway, a correlation network analysis was performed in three species. A correlation analysis was performed using the OmicStudio tools at https://www.omicstudio.cn/tool/62, accessed on 3 March 2022. The correlation between genes was tested using the average of their expression at different time points. Paired Pearson correlation coefficients were calculated between all genes in each subset. To avoid choosing arbitrary numerical correlation thresholds, we applied the context likelihood of relatedness algorithm (CLR) to define or reject correlations between metabolites. We chose the degree of the node (hereafter degree) as the network topology parameter, which allowed us to select the node with the highest number of interactions in the network. The degree of a node is defined by the number of edges that link it to other nodes in the network. In our current study, the nodes with the highest interaction (the nodes with the highest number of connections) represent the key genes of the pathway.

### 4.7. Statistical Analysis of Data

All data were statistically analyzed using GraphPad Prism and SPSS software. For multiple comparisons, the data firstly was performed by one-way analysis of variance (ANOVA) to check the equality of variance (Levene-test), then the Tukey’s multiple comparison tests was used to determine the significant difference (*p* < 0.05) of means with the SPSS software. All the assays described above were repeated at least three times on three biological replicates.

## 5. Conclusions

In our study, 4.8-DHT treatment affected the leaf and root development of *Digitaria sanguinalis*, *Arabidopsis thaliana,* and *Poa annua*. With the help of transcriptomic data, our research found that since each species is differently sensitive to 4,8-DHT, different species respond differently to 4,8-DHT, with differences in the key pathways and genes that trigger it. However, it is possible to further focus on the site of action of 4,8-DHT in photosynthesis and oxidative phosphorylation. Simultaneously, glutathione metabolism and phenylpropanoid-biosynthesis-pathway-related antioxidant enzyme genes also up- and downregulate the response to 4,8-DHT, affecting ROS synthesis and scavenging and making plants resistant or exacerbating damage. In conclusion, this study improved our understanding of weed response mechanisms to 4,8-DHT treatment and provided a solid foundation for exploring the mechanisms of 4,8-DHT stress and producing an ecologically benign weed inhibitor.

## Figures and Tables

**Figure 1 plants-12-02728-f001:**
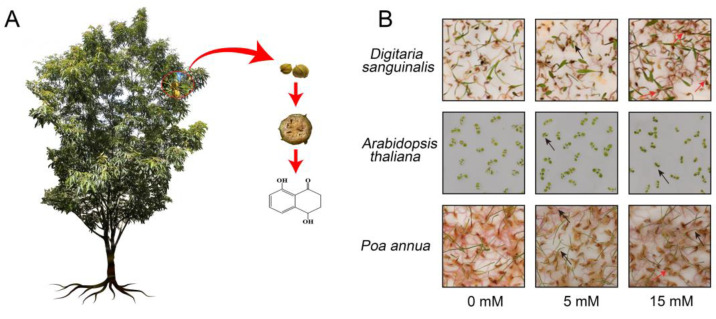
The source and chemical structure of 4,8-DHT (4,8-dihydroxy-1-tetralone) and its effects on the growth of *Digitaria sanguinalis (Ds)*, *Arabidopsis thaliana (At),* and *Poa annua (Pa)*. (**A**) 4,8-DHT was extracted from the outer bark of *Carya cathayensis*. (**B**) The phenotype of *Pa*, *At,* and *Ds* after the treatment of 4,8-DHT. The 7-day old seedlings of *Pa*, *At,* and *Ds* were treated with 0 mM, 5 mM, and 15 mM 4,8-DHT solution for 2 days, respectively. The red arrows represent dead plants at this concentration of 4,8-DHT and the black arrows represent shrunken and curled plants at this concentration of 4,8-DHT.

**Figure 2 plants-12-02728-f002:**
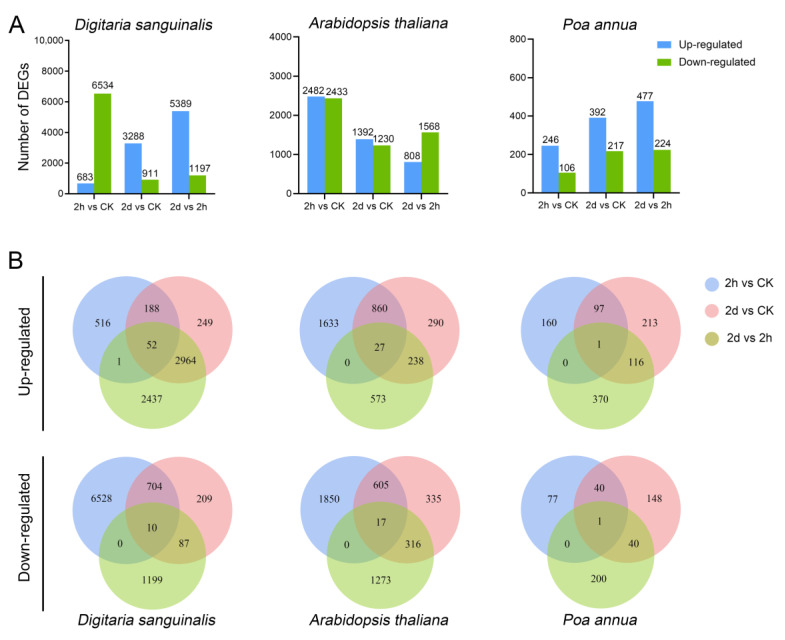
The differentially expressed gene (DEG) summary after 4, 8-DHT treatment in different species. (**A**) Up- and downregulated DEGs in each comparison group: 2 h vs. CK, 2 d vs. CK, and 2 d vs. 2 h for *Digitaria sanguinalis*, *Arabidopsis thaliana,* and *Poa annua*. (**B**) Venn plots of up- and downregulated DEGs in 2 h vs. CK, 2 d vs. CK, and 2 d vs. 2 h for *Digitaria sanguinalis*, *Arabidopsis thaliana,* and *Poa annua*. The number of DEGs exclusively expressed in one sample is shown in each circle of the Venn diagram. The number of DEGs with a common tendency for expression changes between the three comparison groups is shown in the overlapping regions (2 h—the 2 h treatment of 5 mM 4, 8-DHT; 2 d—the 2-day treatment of 5 mM 4, 8-DHT; and CK—the clear water treatment instead of 4, 8-DHT treatment).

**Figure 3 plants-12-02728-f003:**
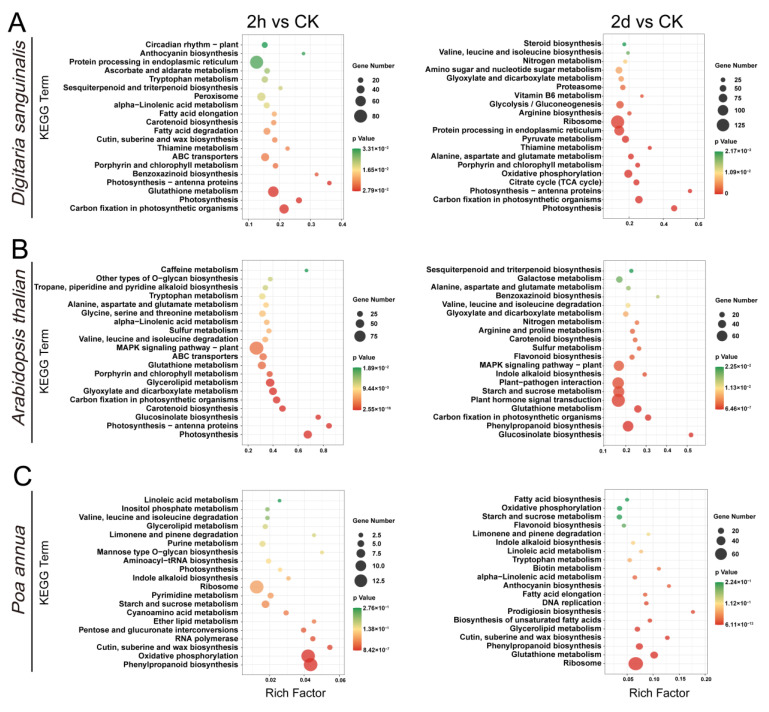
The top 20 KEGG pathways for *Digitaria sanguinalis* (**A**), *Arabidopsis thaliana* (**B**), and *Poa annua* (**C**) at the 2 h vs. CK and 2 d vs. CK after the treatment of 5 mM 4,8-DHT. The *Y*-axis indicates pathway names, and the *X*-axis indicates enrichment factors. The size of the dots indicates the number of DEGs in the pathway and the color of the dots corresponds to different ranges of *q*-values (corrected *p*-values).

**Figure 4 plants-12-02728-f004:**
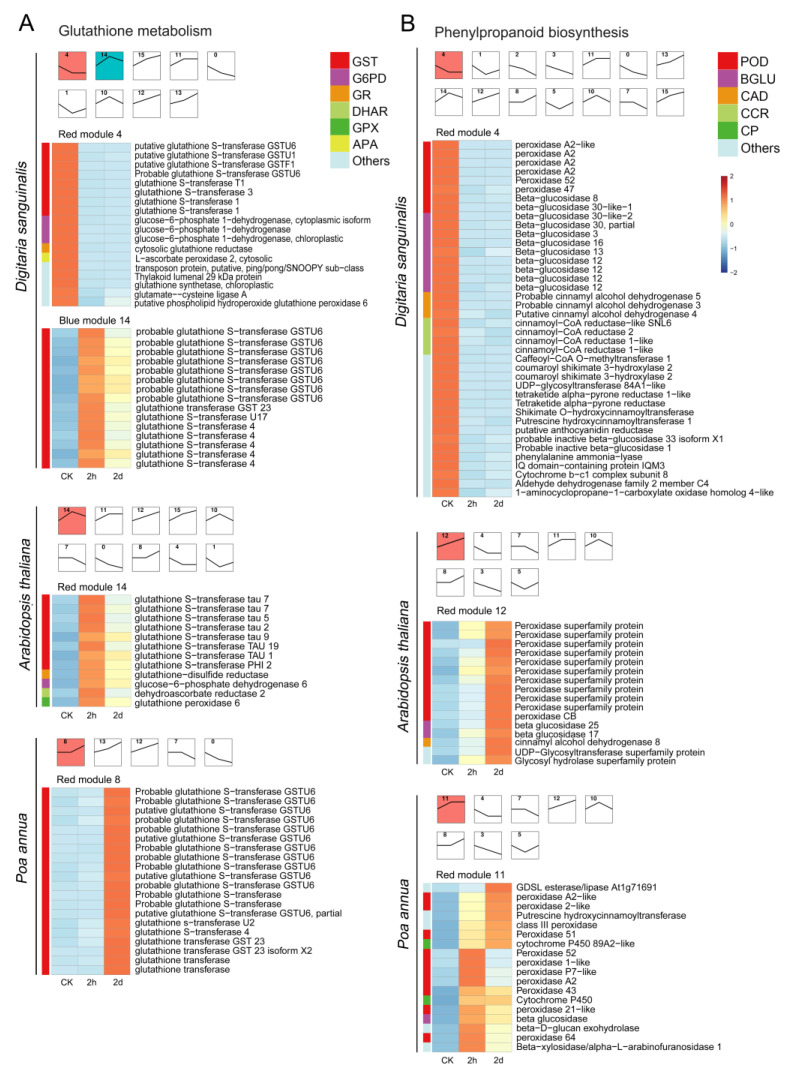
STEM plots and heatmaps of DEGs associated with the anti-ROS pathways after 0 h, 2 h, and 2 d of treatment of 5 mM 4,8-DHT in *Digitaria sanguinalis*, *Arabidopsis thaliana,* and *Poa annua*. (**A**). STEM plots and heatmaps of DEGs involved in glutathione metabolism; (**B**). STEM plots and heatmaps of DEGs involved in phenylpropanoid biosynthesis. STEM has multiple temporal expression patterns, and the colored modules represent the most significant among these DEGs. The different-colored squares on the left of the heatmaps correspond to the genes of the same-colored squares on the right.

**Figure 5 plants-12-02728-f005:**
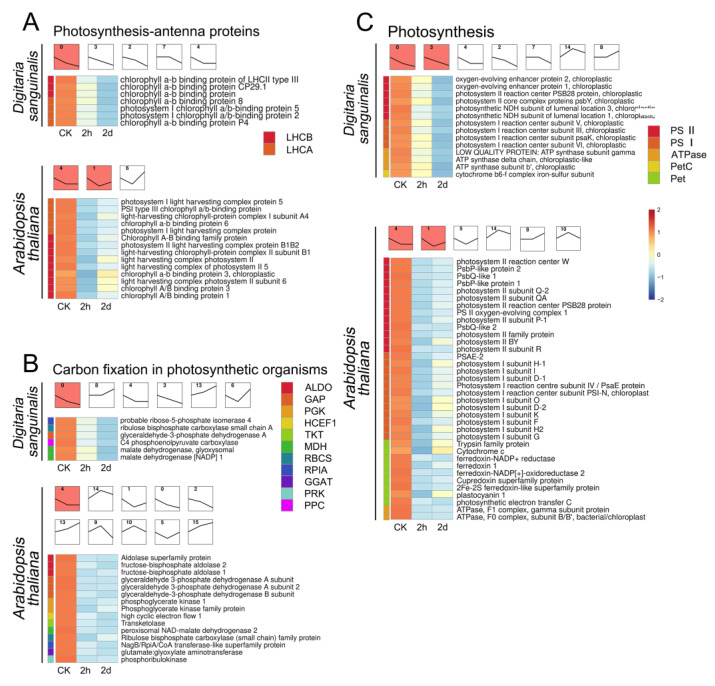
STEM plots and heatmaps of DEGs in photosynthesis-related pathways of *Digitaria sanguinalis* (*Ds*) and *Arabidopsis thaliana* (*At*) after 0 h, 2 h, and 2 d of treatment with 5 mM 4,8-DHT. (**A**) STEM plots and heatmaps of DEGs which are involved in the photosynthesis antenna proteins. (**B**) STEM plots and heatmaps of DEGs are involved in the carbon fixation in photosynthetic organisms. (**C**) STEM plots and heatmaps of DEGs which are involved in photosynthesis. STEM has multiple temporal expression patterns, and the colored modules represent the most significant among these DEGs. The different-colored squares on the left of the heatmaps correspond to the genes of the same-colored squares on the right. The different number in the square only indicates the serial number of the expression patterns.

**Figure 6 plants-12-02728-f006:**
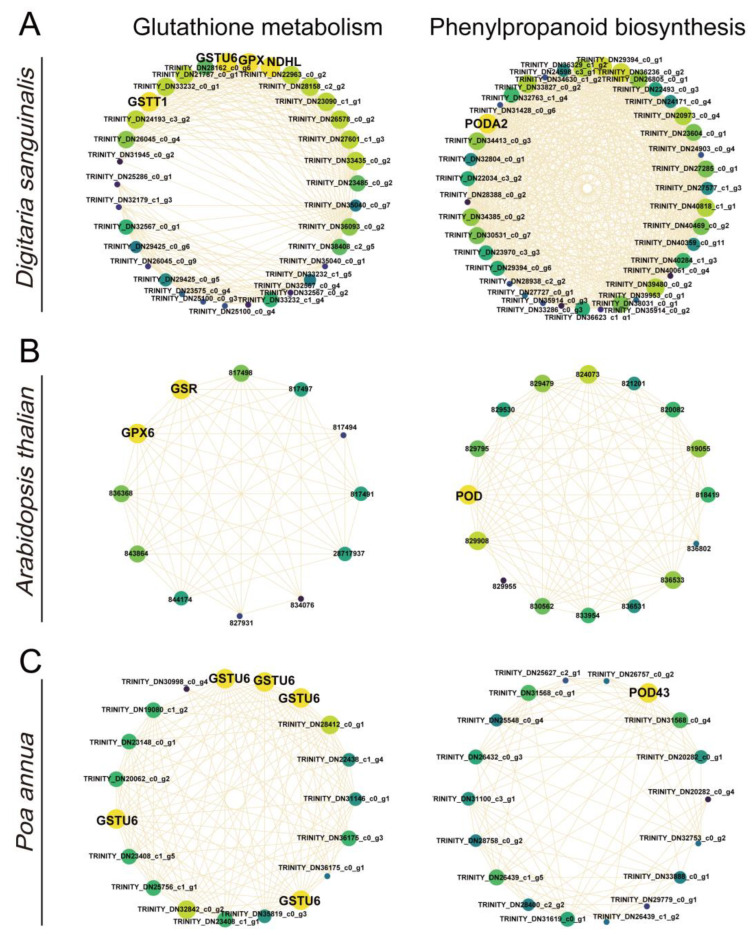
Correlation networks for glutathione metabolism and phenylpropanoid biosynthesis *Digitaria sanguinalis* (**A**), *Arabidopsis thaliana* (**B**), and *Poa annua* (**C**). Yellow represents key genes, and genes with higher linkage counts in the corresponding networks are shown in the larger circles.

**Figure 7 plants-12-02728-f007:**
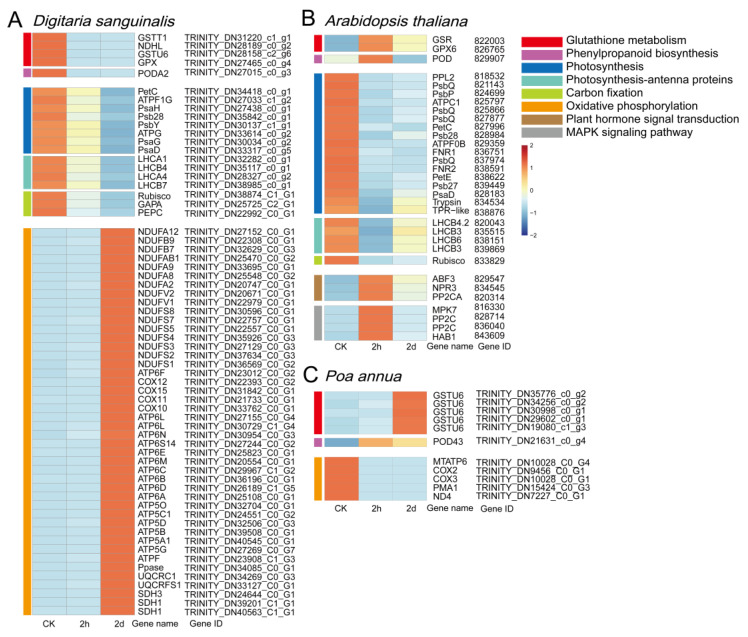
Heat maps of key genes for *Digitaria sanguinalis* (**A**), *Arabidopsis thaliana* (**B**), and *Poa annua* (**C**) after the treatment with 5 mM 4,8-DHT. The color entries on the left of the heat maps represent the pathways in which key genes are located.

**Figure 8 plants-12-02728-f008:**
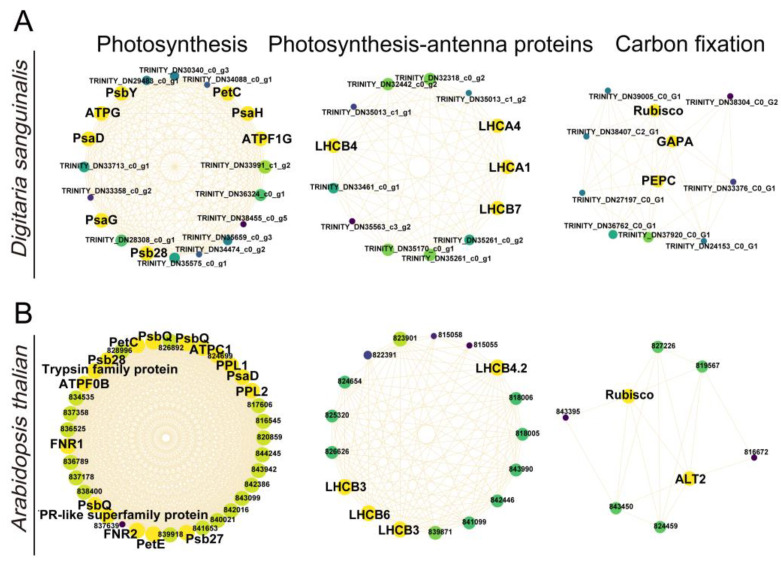
Correlation networks for the photosynthesis, photosynthesis antenna proteins, and carbon fixation in *Digitaria sanguinalis* (**A**) and *Arabidopsis thaliana* (**B**). Yellow represents key genes, and genes with higher linkage counts in the corresponding networks are shown in the larger circles.

**Figure 9 plants-12-02728-f009:**
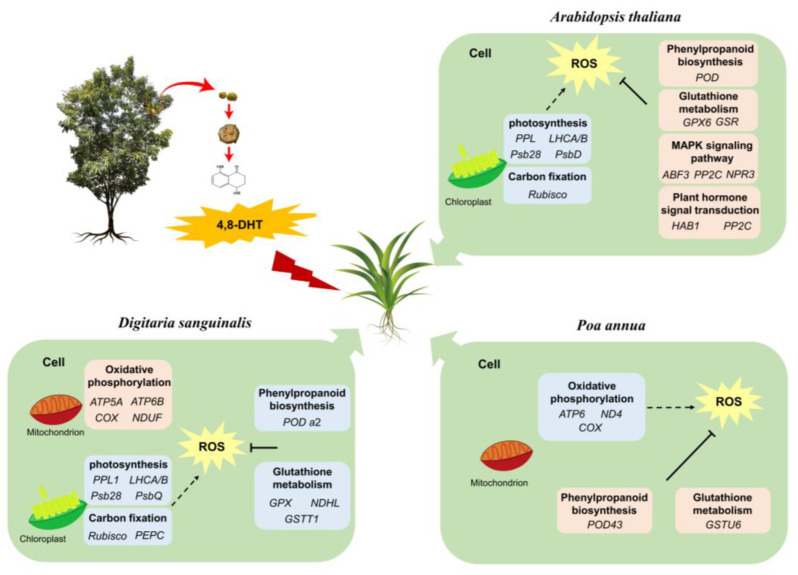
Models of the responses of *Digitaria sanguinalis*, *Arabidopsis thaliana,* and *Poa annua* to 4,8-DHT toxicity. Genes in blue frame represent downregulated expression at 2 h or 2 d; genes in pink circles represent upregulated expression at 2 h or 2 d.

## Data Availability

Not applicable.

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
