# Peer review of "Transcriptome Analysis Reveals the Response Mechanism of Digitaria sanguinalis, Arabidopsis thaliana and Poa annua under 4,8-Dihydroxy-1-tetralone Treatment"

_plants, 2023, doi:10.3390/plants12142728_

Round 1

Reviewer 1 Report

This paper contains many results by laborious and expensive research. But the concentration use are too much as herbicide, selected plants are not proper, as two Gramineae with different, contradictory response, better to use 100 to 1000 times lower concentrations and other family such as Leguminosae.  

Reviewer 2 Report

Dear authors,

I had the opportunity to review the manuscript entitled "Transcriptome analysis reveals the response mechanism of Digitaria sanguinalis, Arabidopsis thaliana, and Poa annua under 4,8-dihydroxy-1-tetralone treatment." The primary objective of this study is to investigate the effects of the DHT molecule on three different plants, with a particular focus on the transcriptome. I find the approach very interesting, especially in the field of allelopathy. However, I have some concerns regarding certain aspects of the experimental design and conclusions.

One major concern I have is the concentrations of DHT used in the study. The concentrations of 5 and 15 mM are quite high, especially when considering allelochemicals. Additionally, since it is not clear how this DHT molecule was extracted and in what concentration/quantity, these values might be out of range, potentially exaggerating the responses of the studied plants. Could the authors address this concern comprehensively?

Here is a list of my comments:

- Starting with the introduction, I believe that the paragraph from L56 to L77 could be either avoided or at least reduced, as it appears somewhat repetitive with the subsequent discussion.

- L87, also known as hormesis, is a phenomenon quite common in other herbicides, such as glyphosate. However, demonstrating it is not as straightforward or simple, although it may seem so at first glance.

- L91, could DHT be legitimately defined as an herbicide? Since the Mode of Action is unknown?

- L92, why specifically these three plants? Arabidopsis is understandable, but what is the reason for selecting the other two? If there is a specific rationale, it would be better to integrate it into the introduction or discussion.

- L99-112, this part raises the most perplexity. There is much discussion about reduced germination or growth reduction, but all that is shown are (often unclear) photos of germinated seeds. Where are the data regarding phenotypic analyses? Based on what parameters were the plants chosen for transcriptomics? Please provide justification and integration.

- L114, was the DHT extracted or purchased? If it was extracted, how was it done, as it is not specified elsewhere? Please provide motivation.

- Fig3, the text in the various plots is very small.

- Fig4, same issue.

- Fig5, same issue.

- Fig8, same issue.

- L460, obtained how?

- L465, it is not clear how the plants were actually selected for the analyses. Please clarify.

From time to time some sentences should be clearer.

Round 2

Reviewer 1 Report

I recommend you to reduce the concentarions.